# New Method of Cardiac Lead Evaluation Using Chest Radiography

**DOI:** 10.3390/medicina58020222

**Published:** 2022-02-01

**Authors:** Rafal Mlynarski, Sebastian Glowoc, Agnieszka Mlynarska, Krzysztof Golba

**Affiliations:** 1Department of Electroradiology, School of Health Sciences, Medical University of Silesia, 40-635 Katowice, Poland; joker@mp.pl; 2Department of Electrocardiology, Upper Silesian Medical Centre, 40-635 Katowice, Poland; sebastianglowoc@gmail.com (S.G.); kgolba@sum.edu.pl (K.G.); 3Department Gerontology and Geriatric Nursing, School of Health Sciences, Medical University of Silesia, 40-635 Katowice, Poland; 4Department of Electrocardiology and Heart Failure, School of Health Sciences, Medical University of Silesia, 40-635 Katowice, Poland

**Keywords:** X-ray image, chest radiography, cardiac pacemaker, complications

## Abstract

*Background and Objectives*: There is currently no method that can be used for a precise evaluation of pacemaker leads using X-ray images, which could be a valuable add-on in the follow-up of patients. The aim of this paper was to create a simple method to measure selected points and lines using the chest radiography of patients with cardiac pacemakers. *Materials and Methods*: The study included 99 patients after permanent pacemaker implantation (72.0 ± 12.9 y; 58 W). The newly created method was used to evaluate the course of the leads based on an X-ray of the chest in an AP (posterior–anterior) projection (standing up) with optimization. The chest X-ray was applied to the original measurement grid, which was determined by specific anatomical points. For the purpose of this paper, a number of measurable parameters have been proposed. *Results*: The technical quality of the images was very good (4.50 ± 0.72). There were no gender-specific differences: women 4.46 ± 0.75/men 4.56 ± 0.67; *p* = 0.5183. The quality of the imaging of the leads was also good (3.72 ± 0.83), and no statistical differences were found between the genders. After verifying the technical quality of the X-ray images, the tract of the leads was measured. The only significant difference was found in parameter E—this value describing the so-called “death bend”, which was significantly lower in women (3.98 ± 1.35) vs. men (4.58 ± 1.49): *p* = 0.039. *Conclusions*: The presented method permitted the leads of a cardiac pacemaker to be precisely described with good clinical validity using chest radiography.

## 1. Introduction

An artificial pacemaker is a device that is used for the electrical correction of the heart beat [1,2]. The device is implanted subcutaneously, usually below the left clavicle. The lead or leads are inserted into the heart through the venous vessels under radiological control. Depending on the expected type of pacing, the device is implanted into the right ventricle, right atrium or both of the heart cavities. During the procedure, using the intra-cardiac signal that is obtained from the leads, the resistance and the pacing threshold are measured. These measurements enable the operator/surgeon to confirm the accuracy of implantation. The type of pacemaker and the type of pacing that are used depend on the reason for the pacemaker implantation. Currently, the guidelines of the European Society of Cardiology are the basis for cardiac pacemaker implantation. In the vast majority of cases, the implantation of these classical anti-arrhythmic systems proceeds without any complications and takes about 60 min [3]. In a small percentage of cases, complications may occur during surgery or sometimes after it because the implantation of a cardiac pacemaker is fraught with unlikely but serious risks. These include early complications such as the formation of a hematoma at the pacemaker site, the occurrence of pneumothorax or the dislocation of the leads in the heart. Late infectious complications should also be mentioned. One of the leading methods for diagnosing and monitoring the treatment of the above-mentioned complications are imaging methods [4]. At present, X-ray diagnostics is the preferred method due to its dissemination, acceptable quality and, most importantly, the safety of the patient and the implanted devices [5]. 

There are no methods/protocols for the precise evaluation of pacemaker leads using X-ray images, which could be a valuable add-on for the follow-up of those patients including potential complications from treatment. We attempted to create a simple method to measure selected points and lines from the chest radiography of patients with cardiac pacemakers, as well as a virtual model that could facilitate the further analysis of the resulting images.

## 2. Materials and Methods

This study included 99 consecutive patients who had a permanent pacemaker implantation (72.0 ± 12.9 years, 58 women) and were hospitalized in the Department of Electrocardiology of the Independent Public Clinical Hospital No. 7 of the Medical University of Silesia in Katowice. All of the devices were implanted by experienced operators who perform more than 75 procedures per year.

For all of the patients who were included into the study, chest radiography was performed for clinical reasons (exact information is provided in the inclusion criteria below). A formal response from the Bioethical Committee of the Medical University of Silesia was received and information that the above-mentioned project did not require consent was obtained (KNW/0022/KB/129/17).

### 2.1. Inclusion Criteria

Patients with an implanted single- (VVI type) or dual-chamber (DDD type) cardiac pacemaker who had a chest X-ray performed for clinical reasons were included.

The following criteria were defined as the clinical reasons for this:-Assessment of the position of the leads for determining lead dislocations;-Looking for any damage;-Planning for the next electrocardiology procedure;-A current or threatening complication;-Diagnosing a pneumothorax;-Diagnosing a perforation and other analyses.

### 2.2. Exclusion Criteria

-Patients with chest radiography images performed in positions other than those desired, e.g., in a prone position;-Patients with X-ray images whose technical quality was considered to be insufficient for analysis;-Patients with other than standard cardiac pacemaker devices implanted such as a cardioverter-defibrillator (ICD) or cardiac resynchronization (CRT);-Patients with implanted leads whose thickness exceeded 9 Fr;-Patients with discarded leads whose presence could affect the image of the leads that were selected for analysis.

### 2.3. Image Analysis

The newly created method that was used to evaluate the course of the intra-cardiac leads was based on an X-ray of the chest in a PA (posterior–anterior) projection (standing up).

All of the images that were analyzed were taken on a stationary X-ray device from Philips Bucky Diagnostics (Best, Eindhoven, The Netherlands) located in the Department of Diagnostic Imaging of the Upper Silesian Medical Center in Katowice. Image analysis was performed based on raw DICOM digital data on an eFilm Workstation 3.1 (Merge Healthcare, Chicago, IL, USA) Version 3.1.0 (Build 21).

The chest X-rays were standardized using the hard 110–120 kV method, a short exposure time of 0.1–0.2 ms, which enabled the number of artifacts that could lead to incorrect measurements to be reduced. Maintaining a 150 cm source to image–receptor distance allowed any magnification or geometric distortion, which could adversely affect the reproducibility of the obtained results, to be avoided. The correct placement/positioning of the patient also allowed any anatomical distortion, which is very important in measurements, to be avoided.

The chest X-ray was then applied to the original measurement grid, which was determined by specific anatomical lines and points, allowing the individual angles and distances of the placed leads to be determined (Figure 1). Definitions of individual lines are given in Table 1. Thanks to the above-mentioned lines, it was possible to determine individual angles and distances of the implanted leads. If the measurement was taken from the lead, the center of the lead structure was considered as the border of the measurement. The angles were determined between two semi-straight lines formed by tangents, the exact method of measurement is shown in Appendix A (Table A1). In some cases (e.g., eta angle) a negative sign was used to illustrate the measurement to the left side of the X-ray image. 

The author’s assessment scale was used to assess the quality of visualization of the X-ray imaging and leads imaging based on a 5-point Likert scale. In this scale, 1 point means image not acceptable for clinical analysis and 5 points mean optimal image quality, where all anatomical elements are clearly visible. Two researchers evaluated images, if discrepancies occurred, the final verdict was a consensus between them. 

### 2.4. Measurements

For the objectives of this paper, a number of measurable parameters were proposed that in our opinion may have clinical significance. All of the measurements are given in cm. The angle measurements are given in degrees. The analysis of quantitative variables was performed by calculating the mean, standard deviation, median, quartile, minimum and maximum. 

### 2.5. Statistical Analysis

The analysis of qualitative variables was performed by calculating the number and percentage of occurrences of each value. The values of the quantitative variables in the two groups were compared using the Student’s *t* test (when the variable had a normal distribution in these groups) or the Mann–Whitney test (otherwise). The comparability of the measurements was assessed using the inter-class correlation coefficient (ICC—intra-class correlation coefficient).

The level of compliance was interpreted according to the following scheme:ICC below 0.4—poor complianceICC 0.4–0.6—average complianceICC 0.6–0.75—high complianceICC 0.75–1—very high compliance

The variable measurement was assessed by one competent judge (a PhD student; twice, in a time interval) and two different competent judges at the same time (PhD student and a trained radiologist).

The normality of the distribution of the variables was determined using the Shapiro–Wilk test. The significance level of 0.05 was assumed in the analysis. *p* values below 0.05 were interpreted as indicating significant dependencies. The analysis was carried out in the R program, version 3.5.1 [6].

## 3. Results

The characteristics of the patients which were included are presented in Table 2. The first stage of the project was to assess the technical quality of the X-ray images and the quality of the lead imaging on the X-ray. The technical quality of the images was considered to be very good and useful for clinical analyses (4.50 ± 0.72). The quality of the images of leads on the X-ray was considered to be good (3.72 ± 0.83) and useful for clinical analyses. After verifying the technical quality of X-ray images, the course of the lead(s) was measured using the methodology presented earlier.

The results of measurements that were selected for the analysis of the parameters are presented in Table 3. The above analysis was also performed considering gender division; the exact results are presented in Table 4. The only significant difference was found in parameter E; this is the place where two brachiocephalic veins merge on the posterior surface of the first right rib cartilage into the superior vena cava. Parameter E was significantly lower in women: 3.98 ± 1.35 vs. 4.58 ± 1.49; *p* = 0.039. Based on the aforementioned measured values, a normative model of the course of the leads was created, which is presented in Figure 2. The green color indicates the average tract of the intra-cardiac leads for the entire examined group.

### Evaluation of the Reliability/Repeatability of the Measurements

In the case of original methods, it is necessary to verify the reliability/repeatability of the measurements. The exact results are presented in Table 4. This table shows the class-related correlation coefficients (ICC) that were used to compare the assessments that were made by two independent specialists (inter-rater variability) and by the same specialist in the interval of time (intra-rater variability). The ICC values were fully accepted in most cases, thus indicating the high or very high compliance of measurements. The only exceptions were for the evaluation of the image, the visualization of the leads and the measurement of the angle of the eta.

## 4. Discussion

X-ray modality, including chest X-rays (radiography), is a unique imaging method because it is the only imaging method that permits the physical features of antiarrhythmic devices such as cardiac pacemakers and their leads to be evaluated. Knowledge about the normal and abnormal radiographic appearances of these devices and their mechanical and electrical components is crucial and is a key element in the cooperation between radiologists and the cardiologists/surgeons who perform the implantation procedures [5]. In non-complicated cases, routine X-rays of the cardiac pacemaker are not performed in most countries. In this area, the paper of Edwards et al., which attempts to answer the question of whether routine chest radiography after permanent cardiac pacemaker implantation is necessary, is interesting [7]. The authors examined 192 chest radiographs that were performed after the implantation of a cardiac pacemaker. The authors concluded that the position of pacemaker leads on chest radiographs that are performed after implantation does not affect the subsequent clinical course. Their study suggests that routine chest radiography is not necessary following an uncomplicated pacing procedure that has satisfactory pacing parameters and a clinically low probability of an iatrogenic pneumothorax. It is difficult to say whether or not this is a necessary or unnecessary practice. Of course, taking images before a procedure is very valuable in the case of different complications. On the other hand, most implantations are performed using X-rays from devices such as c-arms or angiographs; therefore, theoretically basic/reference images should be available. The usefulness of the X-ray images of antiarrhythmic devices is especially important when any complications occur. The most common complications are lead dislocations and connection problems, a pneumothorax, a perforation of the heart muscle or vein or lead damage [8]. Aguilera et al. wrote that a pneumothorax or hemothorax is usually an immediate complication that should be looked for on every chest radiograph that is taken immediately after a procedure. In such a situation, knowledge about radiographic anatomy is the key [5]. However, there is currently a lack of precise analyses of the radiographic anatomy of the chest, including the lead(s) of the pacemaker tract and the final lead position, in the literature on this subject. The newly developed method of measurement introduced here can potentially simplify the description process and facilitate communication among different specialists. The important question is why did we decide on such measurements? Most likely, these measurements, from the many that have been tested, retain a high repeatability in subsequent measurements. The consistent problem is the complicated course of the leads, and the presented ones have the ability to describe such a complex structure. We also found the only significant difference in parameter E; this value describes the so-called “death bend”. The name comes from the easy possibility of the rupture of the vessels in this region using tools used for the extraction of intra-cardiac leads and (due to the anatomical position) the possibility of recognizing this complication too late.

The inter- and intra-rater analyses using the ICC (inter-class correlation coefficient) confirmed the usefulness of this method. Most of the available papers discuss the visualization of antiarrhythmic leads, and they are descriptive studies not research studies. An example is the paper by Steiner et al., in which many aspects of pacemaker visualization were discussed. This is valuable material, especially for radiologists who have no examples of these measurements [9]. Similarly, the paper by Bejvan is a descriptive paper [10]. Both papers are a must read for radiologists who cooperate with electro cardiologists.

In each new method, the potential practical implantations should be presented. In particular, this method may track the changes in lead placement on successive chest radiographs. This can be particularly useful in the diagnosis of electrotherapy complications such as lead dislodgment or lead perforation. Both complications are rare but are still faced by doctors who implant electrotherapy devices. This method may also facilitate the implantation of the leads to the specified position using intra-operational fluoroscopy. Ultimately, this method may facilitate communication between various electrotherapy centers to which the patient will go. We believe that the method presented here could be a significant contribution to the visualization of cardiac pacemakers using chest radiography, especially because the number and types of implanted devices is still growing.

### Study Limitations

The chest radiography method itself generates some errors. The optimization of the images, which was performed in this research, is always necessary, but it is not always routine practice in many imaging centers, mainly because it is time-consuming. Only the posterior–anterior images were used in this research; lateral images might provide additional information. The presented research is a single-center study; research by other centers would enable a definitive conclusion to be drawn. At the moment, this work presents only a description and suggestions for a method. The clinical use of this method will be proposed in the future.

## 5. Conclusions

The presented method allows the leads of a cardiac pacemaker to be precisely described with a good clinical validity using chest radiography. This can be particularly useful in the diagnosis of electrotherapy complications such as lead dislodgment or lead perforation and may also facilitate the implantation of the leads to the specified position.

## Figures and Tables

**Figure 1 medicina-58-00222-f001:**
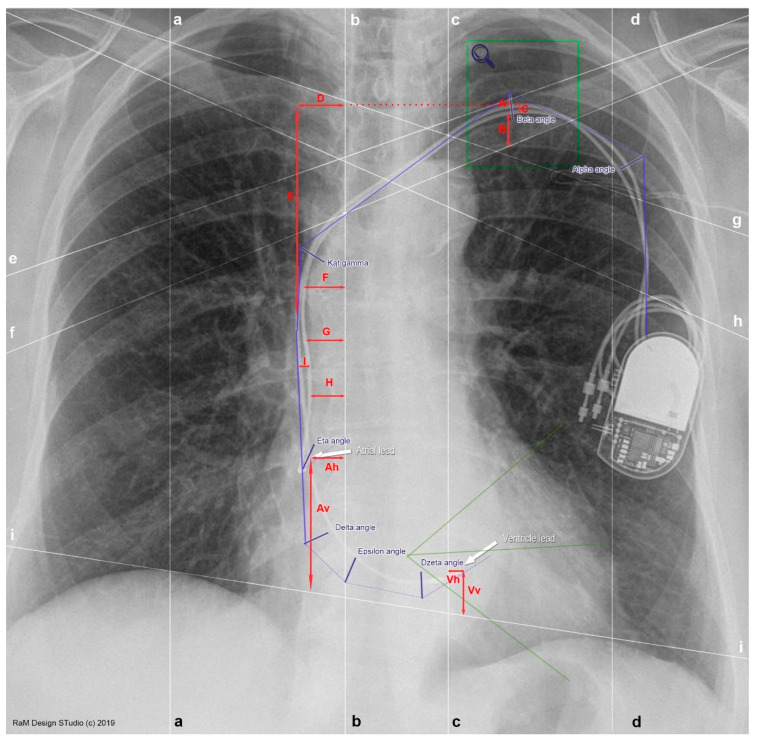
Measurements used in the research based on an example of an X-ray image in a PA (posterior–anterior) projection. (**a**) Midclavicular line—right; (**b**) Lateral sternal line—right; (**c**) Lateral sternal line—left; (**d**) Midclavicular line—left; (**e**) Upper clavicular line—left; (**f**) Lower clavicular line—left; (**g**) Upper clavicular line—right; (**h**) Lower clavicular line—right; (**i**) Diaphragm dome line. Red letters are explained in Appendix A, Table A1.

**Figure 2 medicina-58-00222-f002:**
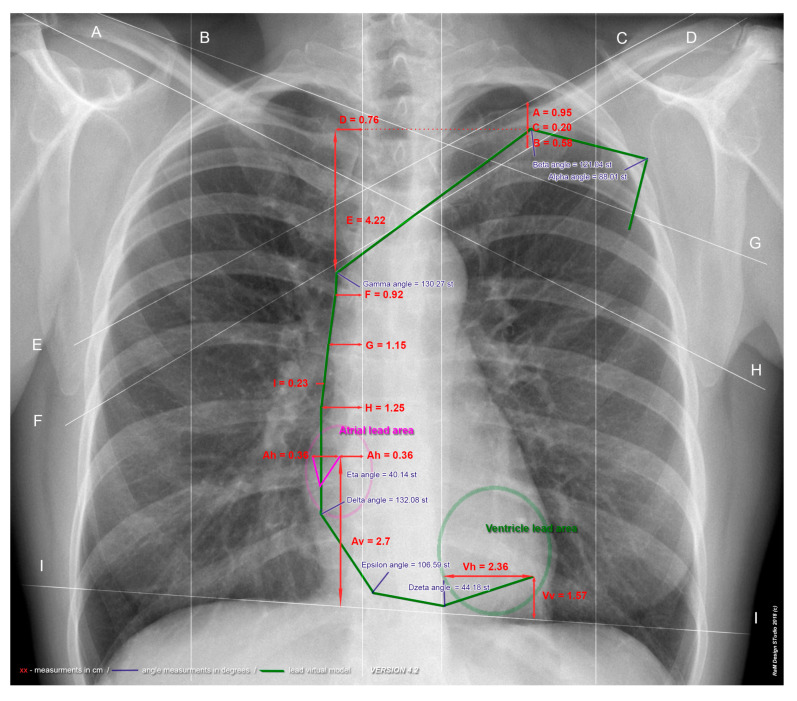
Two-dimensional model of the average course of the leads based on an example of an X-ray image in a PA projection (posterior–anterior). Distance measurements in cm; Angle measurements in degrees. (**A**) Midclavicular line—right; (**B**) Lateral sternal line—right; (**C**) Lateral sternal line—left; (**D**) Midclavicular line—left; (**E**) Upper clavicular line—left; (**F**) Lower clavicular line—left; (**G**) Upper clavicular line—right; (**H**) Lower clavicular line—right; (**I**) Diaphragm dome line. Red letters are explained in Appendix A, Table A1.

**Table 1 medicina-58-00222-t001:** Definitions of the anatomical lines used to make measurements.

**a**	**Midclavicular line—right**
The right midclavicular line runs through the center of the clavicle and then through the right nipple (in women it does not always run through the nipple) and is parallel to the other lines on the chest.	
**b**	**Lateral sternal line—right**
The lateral sternal line runs laterally from the median anterior line along the right lateral edge of the sternum.	
**c**	**Lateral sternal line—left**
The lateral sternal line runs laterally from the median anterior line along the left lateral edge of the sternum.	
**d**	**Midclavicular line—left**
The left midclavicular line runs through the center of the clavicle and then through the right nipple (in women it does not always run through the nipple) and is parallel to the other lines on the chest.	
**e**	**Upper clavicular line—left**
Upper clavicular line—left is the tangent line to the upper, visible surface of the left clavicle passing through the upper part of the sternum end of the clavicle and the conical nodule.	
**f**	**Lower clavicular line—left**
Lower clavicular line—left is the tangent line to the lower, visible surface of the left clavicle passing through the lower part of the sternum, end of the clavicle and the subclavian muscle furrow (if visible).	
**g**	**Upper clavicular line—right**
Upper clavicular line—right is the tangent line to the upper, visible surface of the right clavicle passing through the upper part of the sternum, end of the clavicle and the conical nodule.	
**h**	**Lower clavicular line—right**
Lower clavicular line—right is the tangent line to the lower, visible surface of the right clavicle passing through the lower part of the sternum, end of the clavicle and the subclavian furrow (if visible).	
**i**	**Diaphragm dome line**
A line connecting both diaphragm domes.	

**Table 2 medicina-58-00222-t002:** Characteristics of the patients included.

Feature		Value ± SD
Age (years)	mean ± SD	72.04 ± 13
Weight (kg)	mean ± SD	76.53 ± 14.72
Height (cm)	mean ± SD	164.63 ± 9.1
BMI (kg/m^2^)	mean ± SD	28.29 ± 5.26
Gender	Female	58 (58.59%)
	Male	41 (41.41%)
Main reason for the X-ray examination	Pleural edema	3 (3.03%)
	Pleural edema and lead position	11 (11.11%)
	Procedure planning	80 (80.81%)
	Lead position	3 (3.03%)
	Lack of data	2 (2.02%)
Time since pacemaker implantation	Up to 30 days	77 (77.78%)
	More than 30 days	21 (21.21%)
	Lack of data	1 (1.01%)

Abbreviations: BMI—body mass index, SD—standard deviation.

**Table 3 medicina-58-00222-t003:** Average results of measurements selected for the analysis.

Feature	Values	Feature	Values
Alfa angle (°)	88.01 ± 49.79	F (cm)	0.92 ± 0.52
Beta angle (°)	121.04 ± 23.87	G (cm)	1.15 ± 0.65
A (cm)	0.95 ± 0.49	H (cm)	1.25 ± 0.71
B (cm)	0.58 ± 0.57	I (cm)	0.23 ± 0.21
C (cm)	0.2 ± 0.22	Eta angle (°)	40.14 ± 40.35
D (cm)	0.76 ± 0.37	Ah (cm)	0.36 ± 1.46
E (cm)	4.22 ± 1.43	Av (cm)	2.7 ± 3.06
Gamma angle (°)	130.27 ± 15.78	Delta angle (°)	132.08 ± 14.15
F (cm)	0.92 ± 0.52	Epsilon angle (°)	106.6 ± 56.53

**Table 4 medicina-58-00222-t004:** Evaluation of reliability/repeatability of measurements using ICC.

Parameter/Measurements	Inter-Rater Variability		Intra-Rater Variability	
Alfa angle	0.999	very high agreement	0.99	very high agreement
Beta angle	0.992	very high agreement	0.959	very high agreement
A	0.935	very high agreement	0.922	very high agreement
B	0.827	very high agreement	0.824	very high agreement
C	0.776	very high agreement	0.74	high agreement
D	0.826	very high agreement	0.709	high agreement
E	0.792	very high agreement	0.883	very high agreement
Gamma angle	0.903	very high agreement	0.927	very high agreement
F	0.94	very high agreement	0.946	very high agreement
G	0.941	very high agreement	0.937	very high agreement
H	0.873	very high agreement	0.904	very high agreement
I	0.812	very high agreement	0.858	very high agreement
Eta angle	0.455	average agreement	0.45	average agreement
Ah	0.983	very high agreement	0.986	very high agreement
Av	0.994	very high agreement	0.994	very high agreement
Delta angle	0.979	very high agreement	0.978	very high agreement
Epsilon angle	0.982	very high agreement	0.979	very high agreement
Dzeta angle	0.833	very high agreement	0.854	very high agreement
Vh	0.994	very high agreement	0.995	very high agreement
Vv	0.992	very high agreement	0.988	very high agreement

## Data Availability

The data will be available by contacting the corresponding author.

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
