# Peer review of "New Method of Cardiac Lead Evaluation Using Chest Radiography"

_medicina, 2022, doi:10.3390/medicina58020222_

Round 1

Reviewer 1 Report

This paper suggested a new cardiac lead evaluation method using a chest radiograph.

The following corrections are required.

The method proposed by the author is sufficient for clinical significance, but it is difficult to solve the problem with the method proposed as a basis for supporting it.

First, as a data security issue, analysis data must be collected by an accurate protocol.
Second, it is necessary to suggest a method to increase the reliability of quantitative evaluation rather than qualitative evaluation in the variables used for statistical analysis.
Third, in terms of patient characteristics, gender classification is not significant in the proposed evaluation method.

Also, describe clearly what clinical efficacy is in the conclusions described in this paper.

Author Response

Dear Editor and Reviewers,

Thank you for all of your comments and suggestions on our manuscripts. They were very useful to us, and we hope that this revised manuscript now represents a much better scientific quality.

Below are detailed answers to all of the reviews – in red in the manuscript. We request that our manuscript be considered for publication.

Best regards,

Agnieszka Mlynarska in the name of authors.

---------------------------------------------------------------------------------------------------------------------------

REVIEWER 1

This paper suggested a new cardiac lead evaluation method using a chest radiograph. The following corrections are required.

The method proposed by the author is sufficient for clinical significance, but it is difficult to solve the problem with the method proposed as a basis for supporting it.

ANSWER: Thank you for your opinion which is very valuable for us. Below we explain some problems and marked where corrections were created.

First, as a data security issue, analysis data must be collected by an accurate protocol.

ANSWER: Protocol was approved by bioethical committee which guidelines were used to secure the data; also General Data Protection Regulation rules were used during sensitive data. Protocol designed by us used data from the Table 1 which described anatomical lines used to make measurements. This lines were also used to create parameters. In all cases the same protocol was used. To maximum create data comparable, evaluation of reliability / repeatability of measurements using ICC was presented in the Table 4. As you can see in most of cases very high agreement was obtained, however Eta angle have average agreement.

Second, it is necessary to suggest a method to increase the reliability of quantitative evaluation rather than qualitative evaluation in the variables used for statistical analysis.

ANSWER: We believe that semi-automatic method can improve the data in the future. We evaluate both qualitative and quantitative analysis. First type of data were described in the in the first paragraph of results section, the second one consist of several proposed parameters, which results are presented in the Table 3.

Third, in terms of patient characteristics, gender classification is not significant in the proposed evaluation method.

ANSWER: You all right – this was removed from the manuscript, and the table 3 was facilitated due to removing gender analysis. Page 8 / line 260-267

Also, describe clearly what clinical efficacy is in the conclusions described in this paper.

ANSWER: Conclusion paragraph was updated according your suggestion. Page 9 / Line 281 - 284

Reviewer 2 Report

Thank you very much for the opportunity to review and read the work. The work is an interesting and new approach to a topic that is not widely described in specific literature. Of course, there are some minor errors, e.g. no abbreviations under the tables or no practical implications

Author Response

Dear Editor and Reviewers,

Thank you for all of your comments and suggestions on our manuscripts. They were very useful to us, and we hope that this revised manuscript now represents a much better scientific quality.

Below are detailed answers to all of the reviews – in red in the manuscript. We request that our manuscript be considered for publication.

Best regards,

Agnieszka Mlynarska in the name of authors.

---------------------------------------------------------------------------------------------------------------------------

REVIEWER 2

Thank you very much for the opportunity to review and read the work. The work is an interesting and new approach to a topic that is not widely described in specific literature. Of course, there are some minor errors, e.g. no abbreviations under the tables or no practical implications

ANSWER: Thank you for your kind words. The work has been thoroughly analyzed to improve its quality. The abbreviations under the tables have been added, and at the end of the discussion, the paragraph on practical implications has also been added. Thank you.

Round 2

Reviewer 1 Report

This paper has been submitted with all necessary corrections.